# Bodiniosides S–Y, Seven New Triterpenoid Saponins from *Elsholtzia bodinieri* and Their Anti-Influenza Activities

**DOI:** 10.3390/molecules26216535

**Published:** 2021-10-29

**Authors:** Linyao Yang, Jiangchao Du, Rongtao Li, Fei Yu, Jindong Zhong

**Affiliations:** 1Faculty of Life Science and Technology, Kunming University of Science and Technology, Kunming 650500, China; linyaoyang0808@163.com (L.Y.); dujiangchao2021@163.com (J.D.); rongtaolikm@163.com (R.L.); 2Medical School, Kunming University of Science and Technology, Kunming 650500, China; feiyuz8@163.com

**Keywords:** *Elsholtzia bodinieri*, triterpenoid saponins, anti-influenza virus activities, bodiniosides S–Y

## Abstract

Investigation of the n-BuOH extract of the aerial parts of *Elsholtzia bodinieri* led to the isolation of seven new triterpenoid saponins, Bodiniosides S–Y (**1**–**7**, resp.). Their strictures were elucidated on the basis of spectroscopic techniques, including HSQC, HSBC, and HSQC–TOCSY experiments, together with acid hydrolysis and GC analysis. The anti-influenza activities of compounds **1**–**7** were evaluated against A/WSN/33/2009 (H1N1) virus in MDCK cells. The results showed that compounds **2** and **5** exhibited moderate anti-influenza activities against A/WSN/33/2009 (H1N1), with inhibition rates of 35.33% and 24.08%, respectively.

## 1. Introduction

*Elsholtzia bodinieri* Vaniot (Chinese name “Dongzisu”), belonging to the taxonomically diverse group of the family Labiatae, is a medicinal plant that grows in Yunnan and Guizhou Provinces in China. It is commonly known as “yashuacao” and is used as a traditional Chinese medicine for the treatment of cough, headache, pharyngitis, fever and hepatitis [1]. Previous studies on *E. bodinieri* led to the isolation of triterpenoid saponins [2,3,4,5,6], flavonoid glycosides [7,8], sesquiterpene glycosides [9], clerodane diterpenoid glycosides [10], and phenolic constituents [11] from the aerial parts of this plant. As a continuation of our work, we further systematically investigated the chemical components of the aerial parts of this plant. In our search for secondary metabolites with structural diversity and potential anti-influenza virus activity, seven new triterpenoid saponins, Bodiniosides S–Y (**1**–**7**, resp.), were obtained from *E. bodinieri*. Among them, compounds **2** and **5** exhibited moderate inhibition of influenza virus activities with inhibition rates of 35.33% and 24.08%, respectively. Herein, we report the isolation, structural elucidation and anti-influenza virus activities of the isolated compounds.

## 2. Results

The n-BuOH soluble fraction of the 75% aqueous acetone extract of the aerial parts of *E. bodinieri* was subjected to repeated column chromatography over silica gel, Sephadex LH-20, RP-18, and semipreparative reversed-phase HPLC, eluting with various solvent systems, to afford seven new triterpenoid saponins (Figure 1). The spectrums can be found in Appendix A.

Compound **1** was obtained as a white amorphous powder. Its molecular formula was determined as C_61_H_98_O_28_, according to the [M − H]^−^ peak at *m/z* 1277.6148 in the negative HR-ESI-MS, indicating 13 degrees of unsaturation. It exhibited a UV maximum at 204 nm. The IR spectrum showed the presence of hydroxyl (3441 cm^−1^), carbonyl (1722 cm^−1^), and olefinic (1635 cm^−1^) groups. NMR data analysis indicated that **1** was a saponin containing a triterpene sapogenin and five monosaccharides.

In the ^1^H and ^13^C NMR spectra of aglycone moiety, 6 tertiary methyl groups [*δ*_H_ 0.81, 0.83, 0.84, 0.88, 1.07, and 1.25 (each, 3H, s); *δ*_C_ 12.9, 15.9, 17.5, 23.6, 25.5, and 30.6, (each, q)], 11 methylenes containing an oxygenated one [*δ*_H_ 4.03 (1H, m, H_a_-23), 4.12 (1H, m, H_b_-23; *δ*_C_ 64.7 (t, C-23)], 5 methines including an oxygenated one[*δ*_H_ 3.73 (1H, br. s, H-3); *δ*_C_ 81.9 (d, C-23)] and 1 unsaturated one [*δ*_H_ 5.27 (1H, m, H-12); *δ*_C_ 123.2 (d, C-12)], as well as 8 quaternary carbons (including a carbonyl carbon (*δ*_C_ 176.4, C-28) and a unsaturated one (*δ*_C_ 144.0, C-13)) were observed (Table 1 and Table 2). This information suggested that the aglycone moiety of compound **1** was 3*β*, 23-dihydroxyolean-12-en-28-oic acid [12]. Except for the signals for the aglycone, the remaining 31 signals were assigned as five sugar moieties and an acetoxy group, due to signals of *δ*_C_ 172.7 and 20.9. Moreover, comparison of its ^1^H and ^13^C NMR spectroscopic data with those of Bodinioside H [4], suggested that they had same 3-hydroxy-23-acetoxy-olean-12-en-28-oic acid as the aglycone, but differed in the sugar moiety.

The ^1^H NMR spectrum showed five signals for anomeric protons at *δ*_H_ 6.45 (1H, br. s), 6.09 (1H, d, *J* = 7.8 Hz), 5.16 (1H, d, *J* = 7.8 Hz), 4.97 (1H, d, *J* = 7.8 Hz) and 4.86 (1H, d, *J* = 7.8 Hz), which correlated with anomeric carbon signals at *δ*_C_ 101.0, 94.5, 107.2, 107.1 and 105.3 in the HSQC spectrum, respectively, suggesting the presence of five sugar moieties. Acid hydrolysis of **1** with 1 M HCl produced L-rhamnose (Rha), D-glucose (Glc), and D-xylose (Xyl) as sugar residues by GC chromatography with the corresponding trimethylsilylated L-cysteine derivatives. Since NMR signals of five sugar units have undesirable overlapped effects, the HMQC-TOCSY experiment was successfully used to distinguish and assign the ^1^H and ^13^C NMR signals of each sugar moiety. The correlations from the anomeric proton signal at *δ*_H_ 4.97 to three carbon signals at *δ*_C_ 75.3, 78.4, and 67.1, as well as from three proton signals at *δ*_H_ 4.22, 4.06, and 3.75 to the anomeric carbon, suggested the presence of D-xylopyranose. In a similar way, the ^1^H and ^13^C NMR signals for D-glucopyranosyl and L-rhamnopyranosyl were assigned. In addition, the *J*
_H1, H2_ coupling constants of four anomeric proton signals at *δ*_H_ 6.09 (1H, d, *J* = 7.8 Hz), 5.16 (1H, d, *J* = 7.8 Hz), 4.97 (1H, d, *J* = 7.8 Hz) and 4.86 (1H, d, *J* = 7.8 Hz) suggested the *β* anomeric configuration for the xylopyranosyl and glucopyranosyl units. The inspection of the anomeric proton (6.45, br. s) deduced the *α* anomeric configuration for the L-rhamnopyranosyl unit [4].

The sequence of the glycoside chains connected to C-3 and C-28 was established by analysis of the HMBC correlations (Figure 2). The absence of any glycosidation shift for Xyl suggested that Xyl was the singlet sugar unit attached at C-3 of the aglycone, which was further confirmed by HMBC correlation of H_Xyl_-1 (*δ*_H_ 4.97) of C-3. A series of HMBC correlations from H_Glc_-1 (*δ*_H_ 6.09) to C-28, from H_Rha_-1 (*δ*_H_ 6.45) to C_Glc_-2 (*δ*c 76.3), from H_Glc′_-1 (*δ*_H_ 4.86) to C_Glc_-6 (*δ*c 68.2), and from H_Glc′′_-1 (*δ*_H_ 5.16) to C_Rha_-4 (*δ*c 85.7) enabled the sugar chain of C-28 to be assigned as *β*-D-glucopyranosyl(1→6)-[*β*-D-glucopyranosyl-(1→4)-*α*-L-rhamnopyranosyl(1→2)]-*β*-D-glucopyranoside. Thus, the structure of compound **1** was elucidated to be 3-*O*-*β*-D-xylopyranosyl-23-acetyloxy-olean-12-en-28-oic acid 28-*O*-*β*-D-glucopyranosyl(1→6)-[*β*-D-glucopyranosyl-(1→4)-*α*-L-rhamnopyranosyl(1→2)]-*β*-D-glucopyranoside, a new oleanane triterpenoid saponin, named Bodinioside S.

Compound **2** was obtained as white amorphous powder. Its positive HR-ESI-MS spectrum indicated the molecular formula to be C_60_H_96_O_27_ by the observation quasi-molecular ion peak [M − H]^−^ at *m/z* 1247.6070 and with the help of the NMR spectroscopic date, indicating 13 degrees of unsaturation. Detailed comparison of the ^1^H and ^13^C NMR spectral data (Table 1 and Table 2) of **2** with those of **1** revealed that they were highly structural similar, except for the replacement of signals for the Glc’ at C-28 in **1** by the Ara in **2**. Acid hydrolysis of **2** yielded L-rhamnose, D-glucose, L-arabinose and D-xylose as sugar residues as determined by GC analysis. In the ^1^H NMR spectrum of **2**, five anomeric H-atom at *δ*_H_ 6.47 (br. s), 6.14 (d, *J* = 7.8 Hz), 5.62 (br. s), 5.12 (d, *J* = 7.8 Hz) and 4.86 (d, *J* = 7.5 Hz) correlated with anomeric carbon signals at *δ*_C_ 101.2, 94.4, 109.7, 107.0, and 107.2 in the HSQC spectrum, respectively, suggesting the presence of five sugar residues: one rhamnopyranosyl (Rha), one arabinopyranosyl (Ara), one xylopyranosyl (Xyl) and two glucopyranosyl (Glc and Glc’) units. The Xyl unit was still linked to C-3 (*δ*_C_ 82.8) of the aglycone based on the HMBC correlation between H_Xyl_-1 (*δ*_H_ 4.86) of Xyl and C-3 (*δ*c 82.8). A series of HMBC correlations from H_Glc_-1 (*δ*_H_ 6.14) to C-28, from H_Rha_-1 (*δ*_H_ 6.47) to C_Glc_-2 (*δ*c 76.3), from H_Ara_-1 (*δ*_H_ 5.62) to C_Glc_-6 (*δ*c 68.3), and from H_Glc′_-1 (*δ*_H_ 5.12) to C_Rha_-4 (*δ*c 85.7) enabled the sugar chain of C-28 to be assigned as *α*-L -arabinopyranosyl(1→6)-[*β*-D-glucopyranosyl-(1→4)-*α*-L-rhamnopyranosyl(1→2)]-*β*-D-glucopyranoside. From the foregoing evidence, the structure of **1** was unequivocally determined to be 3-*O*-*β*-D-xylopyranosyl-23-acetyloxy-olean-12-en-28-oic acid 28-*O*-*α*-L-arabinopyranosyl(1→6)-[*β*-D-glucopyranosyl-(1→4)-*α*-L-rhamnopyranosyl-(1→2)]-*β*-D-glucopyranoside, and named Bodinioside T.

Compound **3** was isolated as white amorphous powder. The molecular formula was established as C_55_H_88_O_23_ by positive HR-ESI-MS (*m*/*z* 1139.5614, [M + Na]^+^) and NMR spectral data, indicating 12 degrees of unsaturation. In the ^1^H NMR spectrum of **3**, four anomeric H-atom at *δ*_H_ 6.45 (br. s), 6.15 (d, *J* = 7.8 Hz), 5.13 (d, *J* = 7.8 Hz) and 4.89 (d, *J* = 7.8 Hz) correlated with anomeric carbon signals at *δ*_C_ 101.2, 94.5, 105.5 and 107.1 in the HSQC spectrum, respectively, suggesting the presence of four sugar residues: one rhamnopyranosyl (Rha), one xylopyranosyl (Xyl), and two glucopyranosyl (Glc and Glc’) units. Detailed comparison of the ^1^H and ^13^C NMR spectra of **3** (Table 1 and Table 2) with those of Bodinioside H [4] revealed that they were identical, except for the absence of signals for Xyl moiety on C-3. GC analysis after acid hydrolysis of **3** as the same manner with **1** gave D-glucose, D-xylose, and L-rhamnose in a ratio of 2:1:1. Moreover, a series of HMBC correlations from H_Glc_-1 (*δ*_H_ 6.15) to C-28, from H_Rha_-1 (*δ*_H_ 6.45) to C_Glc_-2 (*δ*c 76.3), from H_Xyl_-1 (*δ*_H_ 5.13) to C_Glc_-6 (*δ*c 68.3), and from H_Glc’_-1 (*δ*_H_ 4.89) to C_Rha_-4 (*δ*c 85.7), adequately illustrated the structure of **3** as 3*β*-hydroxy-23-acetyloxy-olean-12-en-28-oic acid 28-*O*-*β*-D-xylopyranosyl(1→6)-[*β*-D-glucopyranosyl-(1→4)-*α*-L-rhamnopyranosyl-(1→2)]-*β*-D-glucopyranoside, named Bodinioside U.

Compound **4** was obtained as white amorphous powder. It exhibited a quasi-molecular ion peak at *m/z* 1205.5950 [M − H]^−^ in the negative HR-ESI-MS spectrum, suggesting the molecular formula C_58_H_94_O_26_, indicating 12 degrees of unsaturation. Besides one hydroxyl taking the place of an acetoxy group substituent on C-23, most NMR signals (1 and 2) of **4** were nearly identical to those of Bodinioside H [4]. Five anomeric H-atom at *δ*_H_ 6.38 (br. s), 6.11 (d, *J* = 7.9 Hz), 5.15 (d, *J* = 7.6 Hz), 5.02 (d, *J* = 7.4 Hz) and 4.85 (d, *J* = 7.4 Hz) in the ^1^H NMR spectrum were ascribed to D-xylose, D-glucose, and L-rhamnose, respectively, in combination with acid hydrolysis and GC analysis. The Xyl unit was assigned to C-3 (*δ*c 82.1) of the aglycone on the basis of the long-range correlation between H-1 (*δ*_H_ 5.02) of Xyl and C-3. Meanwhile, a series of HMBC correlations from H_Glc_-1 (*δ*_H_ 6.11) to C-28 (*δ*c 176.7), from H_Rha_-1 (*δ*_H_ 6.38) to C_Glc_-2 (*δ*c 77.6), from H_Xyl′_-1 (*δ*_H_ 4.85) to C_Glc_-6 (*δ*c 68.6), and from H_Glc′_-1 (*δ*_H_ 5.15) to C_Rha_-4 (*δ*c 85.6) clarified the linkage of Glc, Xyl and Rha units at C-28 as shown. The structure of compound **4** was, therefore, concluded to be 3-*O*-*β*-D-xylopyranosyl-23-hydroxy-olean-12-en-28-oic acid 28-*O*-*β*-D- xylopyranosyl-(1→6)-[*β*-D-glucopyranosyl-(1→4)-*α*-L-rhamnopyranosyl-(1→2)]-*β*-D- glucopyranoside, and named Bodinioside V.

The molecular formula of Bodinioside W (**5**) was established as C_52_H_84_O_21_ on the basis of the negative HR-ESI-MS from the quasi molecular ion peak at *m/z* 1089.5454 [M + COOH]^−^, indicating 11 degrees of unsaturation. In the ^1^H NMR spectrum of **5**, four anomeric H-atom at *δ*_H_ 6.51 (br. s), 6.11 (d, *J* = 8.0 Hz), 5.01 (d, *J* = 7.8 Hz) and 4.85 (d, *J* = 7.4 Hz) correlated with anomeric carbon signals at *δ*_C_ 101.3, 94.6, 106.7 and 105.4 in the HSQC spectrum, respectively. Acid hydrolysis of **5** yielded two D-xylose (Xyl), L-rhamnose (Rha) and D-glucose (Glc) as sugar residues by GC chromatography. Interpretation of its NMR data (Table 1 and Table 2) revealed that the structure of compound **5** was closely related to compound **4**, except for the presence of an additional Glc at C-28 of **4**. Thus, the structure of compound **5** was 3-*O*-*β*-D-xylopyranosyl-23-hydroxy-olean-12-en-28-oic acid 28-*O*-*β*-D-xylopyranosyl- (1→6)-[*α*-L-rhamnopyranosyl-(1→2)]-*β*-D-glucopyranoside, and named Bodinioside W.

Compound **6** was obtained as white amorphous powder. It exhibited a quasi-molecular ion peak at *m/z* 835.4459 [M + Na]^+^ in the positive HR-ESI-MS spectrum, suggesting the molecular formula of C_42_H_68_O_15_, indicating nine degrees of unsaturation. The ^1^H NMR spectrum (Table 1 and Table 2) of compound **6** showed two signals for anomeric protons at *δ*_H_ 6.60 (1H, br. s) and 6.23 (1H, d, *J* = 8.1 Hz), which correlated with anomeric carbon signals at *δ*_C_ 101.6 and 95.1 in the HSQC spectrum, respectively, suggesting the presence of two sugar moieties. Acid hydrolysis of **6** yielded L-rhamnose (Rha) and D-glucose (Glc) as sugar residues by GC chromatography. The NMR data of **6** were highly analogous to the sericoside [13], suggested that they had same 2, 3, 19, 23-tetrahydroxy-olean-12-en-28-oic acid as the aglycone, except for the presence of an additional Rha at C-28 of **6**. The HMBC correlations from H_Glc_-1 (*δ*_H_ 6.23) to C-28, and from H_Rha_-1 (*δ*_H_ 6.60) to C_Glc_-2 (*δ*c 75.7), adequately illustrated the structure of **6** as 2*α*, 3*β*, 19*α*, 23- tetrahydroxy-olean-12-en-28-oic acid 28-*O*-*α*-L-rhamnopyranosyl-(1→2)-*β*-D- glucopyranoside, named Bodinioside X.

Compound **7** was obtained as white amorphous powder. It exhibited a quasi-molecular ion peak at *m/z* 835.4456 [M + Na]^+^ in the positive HR-ESI-MS spectrum, suggesting a molecular formula C_42_H_68_O_15_, indicating nine degrees of unsaturation. The ^1^H NMR spectrum (Table 1 and Table 2) of **7** revealed six methyl signals at *δ*_H_ 1.13 (3H, s), 1.39 (3H, s), 1.03 (3H, s), 1.19 (3H, s), 1.62 (3H, s), and 1.06 (3H, s) in correlation with carbons at *δ*_C_ 14.2 (C-24), 17.5 (C-25), 17.4 (C-26), 24.1 (C-27), 26.9 (C-29), and 16.6 (C-30) in the HSQC spectrum, respectively. The signal at *δ*_H_ 5.55 (1H, br. s), corresponding to the carbon at *δ*_C_ 128.2 (C-12), coupled with *δ*_C_ 139.2 (C-13) in the ^13^C NMR spectrum. On the basis of the above spectroscopic data, compound **7** was suggested to possess an ursane-12-ene skeleton [6]. Comparison of its NMR spectroscopic data with those of niga-ichigoside F1 [14], suggested that they had the same 2, 3, 19, 23-tetrahydroxy-urs-12-en-28-oic acid as the aglycone. Detailed comparison of ^1^H and ^13^C NMR spectral data of **7** with those of niga-ichigoside F1 indicated that they are highly structurally similar, except for the presence of an additional Rha at C-28 of **7**. Acid hydrolysis of **7** yielded D-glucose (Glc) and L-rhamnose (Rha) as sugar residues by GC chromatography. The structure of compound **7** was, therefore, concluded to be 2*α*, 3*β*, 19*α*, 23-tetrahydroxy-urs-12-en-28-oic acid 28-*O*-*α*-L-rhamnopyranosyl- (1→2)-*β*-D-glucopyranoside, and named Bodinioside Y.

The anti-influenza A virus activity of compounds **1–7** against strain A/WSN/33/2009 was evaluated in MDCK cells, and their cytotoxicity was measured in parallel with the determination of antiviral activity, using oseltamivir as a positive control. It was found that compounds **1–7** displayed no significant cytotoxicity at 50 µM concentration. Then, under this concentration, the in vitro potential anti-influenza A virus effects of all isolates were investigated. The results showed that compounds **2** and **5** exhibited moderate inhibition of influenza virus activities with inhibition rates of 35.33% and 24.08%, while the inhibition rate of the positive control (oseltamivir) was 71.20%.

## 3. Discussion

In summary, seven new triterpenoid saponins, including six oleanane triterpenoid saponins and a ursane one, named Bodiniosides S–Y, were isolated from the aerial parts of *Elsholtzia bodinieri*. Elucidation of their structures was performed based on extensive spectroscopic analyses. The anti-influenza activities of the isolates against A/WSN/33/2009 (H1N1) virus were investigated. The results demonstrated that compounds **2** and **5** exhibited potent inhibition of influenza virus activities, with inhibition rates of 35.33% and 24.08%; meanwhile, compounds **1**, **3**, **4**, **6** and **7** were found to be inactive, with an inhibition rate lower than 10%. A previous study revealed that pentacyclic triterpenoids, including ursane, oleanane, and lupane types, have anti-influenza virus activity [15]. Our results further confirmed that the pentacyclic triterpenoids were active against influenza virus. This investigation should provide valuable information for further understanding of *E. bodinieri*.

## 4. Materials and Methods

### 4.1. General Experimental Procedures

Optical rotations were recorded using a Jasco DIP-370 digital polarimeter (Jasco, Tokyo, Japan). UV spectra were performed on a UV-210A spectrophotometer (Shimadzu, Kyoto, Japan). IR spectra were obtained on a Bio-Rad FtS-135 spectrophotometer (Bio-Rad Laboratories, California, CA, USA) with KBr pellets. The 1D- and 2D NMR spectra were run on Bruker DRX-600 instruments (Bruker BioSpin Group, Rheinstetten, Germany) with TMS as an internal standard. ESI-MS and HR-ESI-MS were measured with an API-Qstar-TOF instrument (Applied Biosystem/MSD Sciex, Concord, ON, Canada). GC analysis was taken on Agilent Technologies HP5890 gas chromatograph (Agilent Technologies Inc., Massy, France) with flame ionization detector. Semi-preparative HPLC was run on an Agilent 1200 liquid chromatograph (Agilent Technologies Inc., Palo Alto, CA, USA) with a ZORBAX SB-C18 (5 A^o^, 9.4 × 250 mm) column. Column chromatography (CC) was carried out on silica gel (200–300 mesh, 80–100 mesh, Qingdao Marine Chemical Factory, Qingdao, China), Diaion HP-20SS (63–150 mm, Mitsubishi Fine Chemical Industries Co., Ltd., Tokyo, Japan), ODS-C_18_ (75 μm, YMC Co., Ltd., Tokyo, Japan), and Sephadex LH-20 (Amersham Biosciences AB, Uppsala, Sweden); thin-layer chromatography (TLC) was monitored by TLC plates (Si gel GF_254_, Qingdao Marine Chemical Factory, Qingdao, China), and spots were visualized by spraying with 5% H_2_SO_4_–EtOH, followed by heating on a hot plate. The purity (>95%) of compounds **1–7** was determined by HPLC.

### 4.2. Materials

The aerial parts of *E. bodinieri* were collected in Yuxi city, Yunnan Province, P. R. China, in May 2016, and identified by Dr. Jindong Zhong. A voucher specimen (KMUST 20160005) was deposited at the Laboratory of Phytochemistry, Faculty of Life Science and Technology, Kunming University of Science and Technology.

### 4.3. Extraction and Isolation

The powered air-dried aerial parts of *E. bodinieri* (15 kg) were extracted with 75% aq. Me_2_CO (3 × 35 L, 24 h, each) at room temperature and filtered. The filtrate was concentrated in vacuo, and the resulting residue was extracted successively with CHCl_3_, AcOEt and *n*-BuOH, respectively.

The *n*-BuOH extract (300.0 g) was separated over macroporous resin CC (Diaion HP-20SS) eluting with MeOH/H_2_O (gradient30, 60, 90, and 100%, each 15 L) to obtain four fractions (*Fr*. A–D). *Fr.* C (eluted with 60% MeOH/H_2_O, 86.5 g) was chromatographed successively over Sephadex LH-20 gel column (20%, 30%, 40%, 50%, 60%, and 100% MeOH/H_2_O, each 8 L) to obtain subfractions *Fr*. C-1–C-6. *Fr.* C-1 (31 g) was isolated by ODS CC (eluted with 10%, 30%, 60%, and 100% MeOH/H_2_O) to obtain subfractions *Fr*. C-1-1–C-1-4. *Fr.* C-1-3 (13 g) was chromatographed over silica gel CC (eluted with CHCl_3_/MeOH 15: 1 to 0: 1) to yield *Fr*. C-1-3-1–C-1-3-5. Compounds **1** (*t_R_* = 15.0 min, 5.6 mg) and **2** (*t_R_* = 20.1 min, 6.2 mg) were purified from *Fr*. C-1-3-4 (167 mg) via semi-preparative HPLC (58% MeOH, 3 mL/min). Compounds **3** (*t_R_* = 17.4 min, 4.3 mg) and **6** (*t_R_* = 22.1 min, 4.8 mg) were purified from *Fr*. C-1-3-3 (103 mg) via semi-preparative HPLC (56% MeOH, 3 mL/min). Compound **4** (*t_R_* = 14.6 min, 5.6 mg) was purified from *Fr*. C-1-3-2 (84 mg) via semi-preparative HPLC (52% MeOH, 3 mL/min). *Fr.* C-1-4 (2.5 g) was chromatographed over silica gel CC (eluted with CHCl_3_/MeOH 10: 1 to 0: 1) to yield *Fr*. C-1-4-1–C-1-4-3. Compounds **5** (*t_R_* = 15.6 min, 4.8 mg) and **7** (*t_R_* = 17.6 min, 7.3 mg) were purified from *Fr*. C-1-4-2 (163 mg) via semi-preparative HPLC (55% MeOH, 3 mL/min).

#### 4.3.1. Bodinioside S (**1**)

White amorphous powder. ${\left[\alpha \right]}_{D}^{26.3}$ = −30.43 (*c* = 0.23, MeOH), IR (KBr): 3441, 2942, 1722, 1635, 1384, 1255, 1045 cm^−1^. UV λ_max_ (MeOH) nm (logε): 204 (2.8). ESI-MS (*neg*.) *m*/*z*: 1277 [M − H]^−^, HR-ESI-MS (*neg*.) *m/z*: 1277.6148 [M − H]^−^, (Calcd. for C_61_H_98_O_28_, 1278.6245). ^1^H and ^13^C NMR: Table 1 and Table 2.

#### 4.3.2. Bodinioside T (**2**)

White amorphous powder. ${\left[\alpha \right]}_{D}^{21.7}$ = −11.95 (*c* = 0.15, MeOH), ESI-MS (*neg*.) *m*/*z*: 1247 [M − H]^−^, HR-ESI-MS (*neg*.) *m/z*: 1247.6070 [M − H]^−^, (Calcd. for C_60_H_96_O_27_, 1248.6139). ^1^H and ^13^C NMR: Table 1 and Table 2.

#### 4.3.3. Bodinioside U (**3**)

White amorphous powder. ${\left[\alpha \right]}_{D}^{21.8}$ = −8.24 (*c* = 0.34, MeOH), ESI-MS (*pos*.) *m*/*z*: 1140 [M + Na]^+^, HR-ESI-MS (*pos*.) *m/z*: 1139.5614 [M + Na]^+^, (Calcd. for C_55_H_88_O_23_, 1116.5716). ^1^H and ^13^C NMR: Table 1 and Table 2.

#### 4.3.4. Bodinioside V (**4**)

White amorphous powder. ${\left[\alpha \right]}_{D}^{26.4}$ = −23.17 (*c* = 0.12, MeOH), ESI-MS (*pos*.) *m*/*z*: 1230 [M + Na]^+^, HR-ESI-MS (*neg*.) *m/z*: 1205.5950 [M − H]^−^, (Calcd. for C_58_H_93_O_26_, 1205.5955). ^1^H and ^13^C NMR: Table 1 and Table 2.

#### 4.3.5. Bodinioside W (**5**)

White amorphous powder. ${\left[\alpha \right]}_{D}^{26.5}$ = −23.01 (*c* = 0.23, MeOH), ESI-MS (*neg*.) *m*/*z*: 1044 [M − H]^−^, HR-ESI-MS (*pos*.) *m/z*: 1089.5454 [M + COOH]^−^, (Calcd. for C_52_H_84_O_21_, 1044.5505). ^1^H and ^13^C NMR: Table 1 and Table 2.

#### 4.3.6. Bodinioside X (**6**)

White amorphous powder. ${\left[\alpha \right]}_{D}^{24.6}$ = −20.43 (*c* = 0.16, MeOH), ESI-MS (*pos*.) *m*/*z*: 835 [M + Na]^+^, HR-ESI-MS (*pos*.) *m/z*: 835.4459 [M + Na]^+^, (Calcd. for C_42_H_68_O_15_, 812.4558). ^1^H and ^13^C NMR: Table 1 and Table 2.

#### 4.3.7. Bodinioside Y (**7**)

White amorphous powder. ${\left[\alpha \right]}_{D}^{24.2}$ = −10.69 (*c* = 0.26, MeOH), ESI-MS (*pos*.) *m*/*z*: 835 [M + Na]^+^, HR-ESI-MS (*pos*.) *m/z*: 835.4456 [M + Na]^+^, (Calcd. for C_42_H_68_O_15_, 812.4558). ^1^H and ^13^C NMR: Table 1 and Table 2.

### 4.4. Acid Hydrolysis for Sugar Analysis

Compounds **1–7** (1.0 mg for each compound) were hydrolyzed with 1 M HCl (0.4 mL) and heated at 90–100 °C for 5 h. The mixture was neutralized by the addition of Amberlite IRA400 (OH^−^ form) and then filtered. The filtrate was dried in vacuo, dissolved in 0.2 mL of pyridine containing *L*-cysteine methyl ester (10 mg/mL) and reacted at 60 °C for 1 h. To this mixture, a solution (0.2 mL) of trimethylsilyl imidazole in pyridine (10 mg/mL) was added, and then heated at 60 °C for 1 h. The final mixture was directly analyzed by GC [30QC2/AC-5 quartz capillary column (30 m × 0.32 mm) with the following conditions: column temperature: 180/280 °C; programmed increase 3 °C/min; carrier gas: N_2_ (1 mL/min); injection and detector temperature: 250 °C; injection volume: 4 μL; split ratio: 1/50]. The authentic samples D- and L-glucose, D- and L-xylose, L-arabinose, and L-rhamnose were treated in the same manner. Under these conditions, the retention times of authentic samples D- and L-glucose, D- and L-xylose, L-arabinose and L-rhamnose were 18.29, 18.87, 13.35, 14.01, 14.30 and 14.97 min, respectively. During our studies, identical retention times observed between the different hydrolysates and authentic standards.

### 4.5. Anti-Influenza Virus Activity

The anti-influenza virus activities of compounds **1**–**7** were evaluated, using influenza strain A/WSN/33/2009 (H1N1). For the inhibitory activity assays, compounds **1**–**7** were dissolved and then diluted with DMSO, using Oseltamivir as a positive control. MDCK cells were seeded into 96-well plates, incubated overnight and infected with influenza virus (MOI ¼ 0.1). The cells were suspended in DMEM supplemented with 1% fetal bovine serum (FBS), containing test compound and 2 mg/mL TPCK-treated trypsin, and a final DMSO concentration of 1% was added in each well. After 40 h of incubation, Cell Titer-Glo reagent was added, and the plates were read, using a plate reader [15]. The inhibition rate was calculated by the following formula: inhibition rate (%) = [1 − (luminescence with compounds − luminescence with compounds and virus)/(luminescence with DMSO − luminescence with DMSO and virus)] × 100%. Assessment of anti-influenza virus activity was performed as described previously [16].

## Figures and Tables

**Figure 1 molecules-26-06535-f001:**
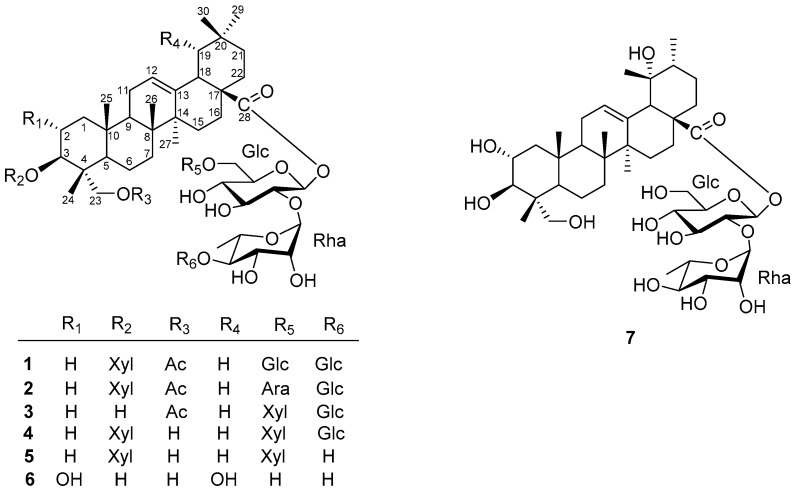
Chemical structure of compounds **1**–**7** from *Elsholtzia bodinieri*.

**Figure 2 molecules-26-06535-f002:**
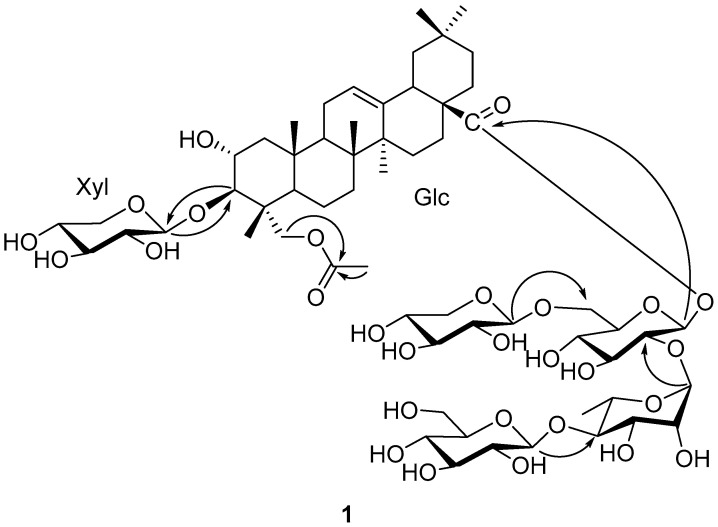
Key HMBC correlation of compound **1**.

**Table 1 molecules-26-06535-t001:** H NMR data of compounds **1**–**7** in Pyridine-*d*_5_ (600 MHz, *δ*_H_ in ppm, *J* in Hz).

Position	1	2	3	4	5	6	7
1	1.57 (m); 0.96 (m)	1.66 (m); 0.98 (m)	1.56 (m); 0.82 (m)	1.49 (m); 1.15 (m)	1.51 (m); 1.03 (m)	1.66 (m); 1.17 (m)	1.66 (m); 1.23 (m)
2	1.81 (m); 1.71 (m)	1.92 (m); 1.76 (m)	1.89 (m); 1.71 (m)	2.20 (m); 2.05 (m)	2.20 (m); 2.01 (m)	4.24 (m)	3.15 (m)
3	3.73 (m)	3.59 (m)	3.90 (m)	4.31 (m)	4.32 (m)	3.41 (m)	3.63 (m)
5	1.52 (m)	1.74 (m)	1.54 (m)	1.69 (m)	1.63 (m)	1.02 (m)	1.80 (m)
6	1.64 (m); 1.35 (m)	1.62 (m); 1.40 (m)	1.73 (m); 1.33 (m)	1.66 (m); 1.37 (m)	1.65 (m); 1.31 (m)	1.78(m); 1.54 (m)	1.68 (m); 1.44 (m)
7	1.82 (m); 1.54 (m)	1.71 (m); 1.55 (m)	1.93 (m); 1.57 (m)	1.87 (m); 1.52 (m)	1.84 (m); 1.50 (m)	1.67 (m); 1.35 (m)	1.73 (m); 1.36 (m)
9	1.66 (m)	1.68 (m)	1.76 (m)	1.70 (m)	1.70 (m)	1.80(m)	1.61 (m)
11	1.93 (m); 1.87 (m)	1.93 (m); 1.82 (m)	1.92 (m); 1.86 (m)	1.97 (m); 1.82 (m)	1.96 (m); 1.84 (m)	2.02(m); 1.92 (m)	1.98 (m); 1.62 (m)
12	5.27 (br. s)	5.42 (br. s)	5.44 (br. s)	5.41 (br. s)	5.40 (br. s)	5.52 (br. s)	5.55 (br. s)
15	1.67 (m); 1.17 (m)	1.57 (m); 1.20 (m)	1.78 (m); 1.21(m)	2.06 (m); 1.56 (m)	2.04 (m); 1.55(m)	2.05 (m); 1.69 (m)	2.03 (m); 1.36 (m)
16	2.03 (m); 1.72 (m)	2.03 (m); 1.68 (m)	2.01 (m); 1.56 (m)	2.20 (m); 2.18 (m)	2.12 (m); 2.07 (m)	2.28 (m); 2.10 (m)	2.24 (m); 2.12 (m)
18	3.12 (dd, 13.6, 3.8)	3.14 (d, 13.5)	3.17 (d, 13.5)	3.13 (d, 13.7)	3.11 (dd, 13.5, 4.0)	2.88 (dd, 13.5, 3.4)	2.87 (s)
19	1.70 (m); 1.17 (m)	2.08 (m); 1.18 (m)	1.89 (m); 1.12 (m)	1.74 (m); 1.15 (m)	1.76 (m); 1.12 (m)	4.25 (m)	
21	1.28 (m); 1.19 (m)	2.20 (m); 1.14 (m)	1.29 (m); 1.01 (m)	1.42 (m); 1.04 (m)	1.30 (m); 0.94 (m)	2.04 (m); 1.72 (m)	2.04 (m); 1.78 (m)
22	1.68 (m); 1.32 (m)	1.39 (m); 1.00 (m)	1.75 (m); 1.35 (m)	1.68 (m); 1.35 (m)	1.64 (m); 1.08 (m)	2.08 (m); 1.78 (overlap)	2.29 (m); 1.42 (m)
23	4.12 (m); 4.03 (m)	4.45 (m); 4.03 (m)	4.30 (m); 4.10 (m)	4.32 (m); 3.63 (m)	4.33 (m); 3.61 (m)	4.36 (m); 3.04 (m)	4.33 (m); 4.02 (m)
24	0.83 (s)	0.84 (s)	0.88 (s)	0.90 (s)	0.87 (s)	1.04 (s)	1.13 (s)
25	1.07 (s)	1.09 (s)	1.14 (s)	1.20 (s)	1.09 (s)	1.09 (s)	1.39 (s)
26	0.81 (s)	0.83 (s)	0.84 (s)	0.82 (s)	0.82 (s)	0.84 (s)	1.03 (s)
27	0.84 (s)	0.89 (s)	0.93 (s)	0.99 (s)	0.90 (s)	1.15 (s)	1.19 (s)
29	1.25 (s)	1.26 (s)	0.97 (s)	1.23 (s)	0.97 (s)	1.54 (s)	1.62 (s)
30	0.88 (s)	0.91 (s)	1.26 (s)	1.48 (s)	1.16 (s)	1.11 (s)	1.06 (s)
23-A_C_O	2.15 (s)	2.14 (s)	2.08 (s)				
3-Xyl							
1	4.97 (d, 7.8)	4.86 (d, 7.8)		5.02 (d, 7.4)	5.01 (d, 7.8)		
2	4.22 (m)	4.20 (m)		4.08 (m)	4.08 (m)		
3	4.06 (m)	4.14 (m)		4.14 (m)	3.99 (t, 8.3)		
4	4.12 (m)	4.35 (m)		4.21 (m)	4.13 (m)		
5	3.75 (m); 3.73 (m)	4.42 (m); 3.74 (m)		4.32 (m); 3.63 (m)	3.64 (m); 3.60 (m)		
28-*O*-sugar Glc							
1	6.09 (d, 7.8)	6.14 (d, 7.8)	6.15 (d, 7.8)	6.11 (d, 7.9)	6.11 (d, 8.0)	6.23 (d, 8.1)	6.16 (d, 7.9)
2	4.27 (m)	4.58 (m)	3.97 (t, 8.1)	4.28 (m)	4.36 (t, 8.3)	4.01 (m)	4.15 (m)
3	4.25 (m)	4.07 (m)	4.05 (m)	4.10 (m)	4.04 (t, 8.8)	4.24 (m)	4.61 (d, 9.8)
4	4.30 (m)	4.54 (m)	3.89 (m)	4.23 (m)	4.23 (m)	4.17 (d, 9.5)	4.02 (d, 9.8)
5	4.14 (m)	4.16 (m)	3.78 (m)	4.18 (m)	4.30 (m)	4.60 (m)	4.32 (m)
6	4.70 (m); 4.28 (m)	4.81 (m); 4.65 (m)	4.64 (dd, 11.4, 2.2); 4.45 (m)	4.43 (d, 10.2); 4.38 (m)	4.62 (d, 11.0); 4.35 (m)	3.64 (dd, 11.5, 5.2); 3.52 (dd, 11.5, 2.0)	4.46 (dd, 10.8, 4.2); 4.24 (d, 10.8)
Rha							
1	6.45 (br. s)	6.47 (br. s)	6.45 (br. s)	6.38 (br. s)	6.51 (br. s)	6.60 (br. s)	6.58 (br. s)
2	4.81 (m)	4.81 (overlap)	4.81 (br. s)	4.81 (br. s)	4.76 (br. s)	4.83 (br. s)	4.81 (br. s)
3	4.64 (overlap)	4.77 (m)	4.50 (m)	4.92 (m)	4.52 (m)	4.52 (m)	4.02 (m)
4	4.48 (m)	4.42 (m)	4.42 (m)	4.45 (m)	4.28 (m)	4.40 (m)	4.15 (d, 9.8)
5	4.39 (m)	4.37 (m)	4.39 (m)	4.37 (m)	4.16 (m)	4.14 (m)	4.61 (m)
6	1.81 (d, 6.1)	1.82 (d, 6.0)	1.82 (d, 6.0)	1.80 (d, 6.0)	1.73 (d, 6.0)	1.78 (d, 6.0)	1.75 (d, 6.0)
Glc’/Ara/Xyl							
1	4.86 (d, 7.8)	6.62 (br. s)	5.13 (d, 7.8)	4.85 (d, 7.4)	4.85 (d, 7.4)		
2	4.02 (m)	4.06 (m)	4.06 (m)	4.26 (m)	3.94 (t, 8.0)		
3	4.08 (m)	4.12 (m)	4.71 (dd, 9.5, 3.2)	4.17 (m)	4.25 (m)		
4	4.17 (m)	4.28 (m)	4.28 (m)	4.02 (m)	4.02 (m)		
5	3.87 (m)	4.45 (m); 3.73 (m)	3.87 (m); 3.76 (m)	3.72 (m); 3.70 (m)	3.65 (m); 3.61 (m)		
6	4.57 (d, 11.2); 4.30 (m)						
Glc’’/Glc’							
1	5.16 (d, 7.8)	5.12 (d, 7.8)	4.89 (d, 7.8)	5.15 (d, 7.6)			
2	4.45 (m)	4.57 (m)	4.09 (m)	3.97 (m)			
3	4.38 (m)	4.02 (m)	4.18 (m)	4.12 (m)			
4	4.30 (m)	4.46 (m)	3.65 (t, 9.5)	4.21 (m)			
5	3.94 (m)	3.98 (m)	4.30 (m)	3.78 (m)			
6	4.65 (d, 11.2); 4.28 (m)	4.81 (d, 11.0); 4.67 (m)	4.64 (dd, 11.4, 2.2); 4.45 (m)	4.42 (d, 11.2); 4.33 (m)			

**Table 2 molecules-26-06535-t002:** C NMR data of compounds **1–7** in pyridine-*d*_5_ (150 MHz, *δ*_C_ in ppm).

Position	1	2	3	4	5	6	7
1	39.8	38.5	38.6	39.2	38.6	47.7	47.9
2	26.0	26.4	26.8	26.2	26.1	68.9	68.8
3	81.9	81.9	78.3	82.1	81.9	78.5	79.9
4	42.4	42.1	42.2	43.6	43.3	43.7	43.5
5	48.3	48.3	48.3	48.0	47.0	48.2	48.0
6	18.5	18.4	18.6	18.4	18.1	18.9	18.6
7	33.7	32.2	32.3	33.2	32.2	33.2	33.3
8	41.8	39.8	41.8	40.1	39.8	40.3	40.5
9	48.2	48.3	48.3	48.3	47.6	48.5	41.7
10	36.7	36.8	37.1	37.1	36.8	38.6	38.3
11	23.6	23.7	23.2	25.1	23.7	24.6	24.2
12	123.2	123.3	123.2	123.1	122.6	123.5	128.2
13	144.0	144.0	143.9	144.2	143.9	144.5	139.2
14	42.1	41.8	42.3	42.3	41.8	42.5	42.2
15	30.6	28.8	28.7	28.4	28.5	28.8	29.4
16	23.1	23.3	23.8	23.7	23.3	24.4	26.1
17	47.1	47.1	48.6	47.5	48.1	46.7	48.0
18	42.1	42.4	42.1	42.3	42.1	45.0	54.6
19	46.2	46.6	46.3	47.0	46.3	81.3	72.5
20	30.3	30.6	30.6	30.8	30.6	29.5	42.2
21	32.9	34.2	34.4	36.7	33.8	35.4	26.7
22	32.1	33.8	33.1	32.9	32.8	32.7	37.4
23	64.7	65.6	65.2	64.3	64.3	66.7	66.6
24	12.9	13.0	12.7	13.7	13.5	14.3	14.2
25	15.9	15.9	16.0	16.4	16.1	17.7	17.5
26	17.5	17.4	17.5	17.7	17.4	17.5	17.4
27	25.5	25.6	25.7	26.1	25.7	28.0	24.1
28	176.4	176.5	176.5	176.7	176.4	177.2	176.8
29	30.6	33.0	33.0	33.2	33.0	33.2	26.9
30	23.6	23.7	23.7	23.7	23.7	24.7	16.6
23-A_C_O	171.2	171.1	171.1				
	20.7	20.7	20.7				
3-Xyl							
1	107.1	107.2		107.2	106.8		
2	75.3	75.3		75.7	75.4		
3	78.4	78.4		78.1	77.8		
4	71.5	71.1		71.3	71.0		
5	67.1	67.1		67.2	66.8		
28-*O*-sugar Glc							
1	94.5	94.4	94.5	94.7	94.6	95.1	94.8
2	76.3	76.3	76.3	77.6	76.2	75.7	75.1
3	78.5	79.2	78.3	78.5	78.4	79.8	78.9
4	71.4	71.5	71.1	71.1	71.8	71.4	72.5
5	77.6	77.3	77.4	77.6	77.9	79.1	78.3
6	68.2	68.3	68.3	68.6	69.7	62.1	62.2
Rha							
1	101.0	101.2	101.2	101.5	101.3	101.4	101.3
2	71.3	71.6	71.6	71.3	70.9	72.2	71.4
3	72.4	72.4	72.3	72.1	72.3	72.4	72.2
4	85.7	85.7	85.7	85.6	74.6	73.8	73.7
5	70.9	71.5	70.9	71.1	71.8	69.7	69.6
6	18.6	18.5	18.7	18.8	18.6	18.6	18.6
Glc’/Ara/Xyl							
1	105.3	109.7	105.5	105.7	105.4		
2	75.3	81.9	74.7	74.9	75.5		
3	78.5	76.3	77.9	78.7	77.8		
4	71.3	86.0	72.4	71.8	72.0		
5	78.2	67.2	66.9	67.2	66.8		
6	62.7						
Glc’’/Glc’							
1	107.2	107.0	107.1	106.7			
2	76.3	76.3	75.8	75.5			
3	79.2	79.2	79.3	78.8			
4	71.0	71.5	71.6	71.8			
5	78.5	78.2	77.4	78.7			
6	62.5	62.7	62.8	62.9			

## Data Availability

Not applicable.

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
