# Peer review of "Bodiniosides S–Y, Seven New Triterpenoid Saponins from *Elsholtzia bodinieri* and Their Anti-Influenza Activities"

_molecules, 2021, doi:10.3390/molecules26216535_

Round 1

Reviewer 1 Report

The manuscript is interesting, I suggest more specific results about the other components different from 2-5 in the anti H1N1 activity as well as clarify the other research discussion 

Author Response

Thanks very much for your comments. compounds 2 and 5 exhibited moderate anti-influenza activities against A/WSN/33/2009 (H1N1), with inhibition rate of 35.33% and 24.08%, meanwhile compounds 1, 3, 4, 6 and 7 were found to be inacctive, with inhibition rate lower than 10%. Thank you for your advice, we added it in the discussion.

Reviewer 2 Report

This manuscript reports the isolation, structure elucidation and antiviral activity of seven new compounds from Elsholtzia bodinieri. The new compounds are structurally derivatives of compounds already isolated from the same plant. The work has been done competently and the structures seem to be determined correctly. It worth publication in this journal.

However, the following points should be taken into consideration:

Compounds 2 and 5 displayed moderate anti-influenza activities. The term “potent” refered to these two compounds should be replaced by “moderate” in the abstract and in all manuscript.

P.2, for compound 1:

- positive HR-ESI-MS need to be changed by negative HR-ESI-MS.

- carbonyl caibon should be corrected by carbonyl carbon

P.7, for compound 4

- the ion peak at m/z 1209.5950 is not correct for the molecular formula given; there is a contradiction with that given P.9 which is the right one.

- m/z should be in italic in all the text and added for compound 5.

P.8,

- antiviral activiral activity should be replaced by antiviral activity

- in vitro should be in italic

-5 should be changed by 5A° in ZORBAX SB-C18 column

P.9,

- was should be replaced by were for the purification of compounds 1 and 2; 3 and 6; 5 and 7.

- the unit for IR data should be added for compound 1

Author Response

Q1: Compounds 2 and 5 displayed moderate anti-influenza activities. The term “potent” refered to these two compounds should be replaced by “moderate” in the abstract and in all manuscript.

A1: Thank you for your advice. In the manuscript, we indicated the “potent” activity to “moderate” activity in all manuscript.

Q2: P.2, for compound 1:

- positive HR-ESI-MS need to be changed by negative HR-ESI-MS.

- carbonyl caibon should be corrected by carbonyl carbon

A2: Thank you very much. These were our negligence. The “positive HR-ESI-MS” has been changed by negative HR-ESI-MS, and the “carbonyl caibon” has been corrected by “carbonyl carbon”.

Q3: P.7, for compound 4

- at m/z 1209.5950 is not correct for the molecular formula given; there is a contradiction with that given P.9 which is the right one.

- m/z should be in italic in all the text and added for compound 5.

A3: The mentioned of ion peak for the molecular formula has been revised in the paper. And the m/z has been in italic in all the text and added for compound 5. These were our negligence, I am sorry.

Q4: P.8,

- antiviral activiral activity should be replaced by antiviral activity

- in vitro should be in italic

-5 should be changed by 5A° in ZORBAX SB-C18 column

A4: In the paper, the “antiviral activiral activity has been replaced by “antiviral activity”, “in vitro” has been in italic, and the “5” has been changed by “5A°” in ZORBAX SB-C18 column. Thank you very much.

Q5: P.9,

- was should be replaced by were for the purification of compounds 1 and 2; 3 and 6; 5 and 7.

- the unit for IR data should be added for compound 1

A5: In the paper, the “was” have been replaced by “were” for the purification of compounds 1 and 2, 3 and 6, 5 and 7. And the unit for IR data has been added for compound 1in the paper.

Reviewer 3 Report

The manuscript reported the isolation of seven new saponins from plant Elsholtzia bodinieri collected at China. In addition, isolated compounds were tested against influenza activity.

Supplementary materials should be provided for transparency. The NMR and HR-MS spectra of seven new compounds, and HPLC chromatogram of sugar analysis should be included.

Author Response

Thanks very much for your comments. Supplementary materials for NMR and HR-MS spectra of the seven new compounds has been added. But supplementary materials for HPLC chromatogram of sugar analysis cannot be provided, because we did not preserve the result of HPLC chromatogram of sugar analysis and we don’t have any samples. I am sorry.

Round 2

Reviewer 3 Report

 I have no other comment.